# Knowledge, Attitude, Behavior Practices and Compliance of Workers Exposed to Respirable Dust in a Zambian Copper Mine

**DOI:** 10.3390/ijerph20186785

**Published:** 2023-09-20

**Authors:** Mwaba Sifanu, Thomas K. Taylor, Kennedy K. Kalebaila, Patrick Hayumbu, Lubinda Nabiwa, Stephanus J. L. Linde

**Affiliations:** 1School of Mathematics and Natural Sciences, Copperbelt University, Kitwe P.O. Box 21692, Zambia; kennedy.kalebaila@cbu.ac.zm (K.K.K.); patrick.hayumbu@cbu.ac.zm (P.H.); lubinda.nabiwa@cbu.ac.zm (L.N.); 2School of Built Environment, Copperbelt University, Kitwe P.O. Box 21692, Zambia; taylortk@cbu.ac.zm; 3Occupational Hygiene and Health Research Initiative, North-West University, Potchefstroom 2520, South Africa; stefan.linde@nwu.ac.za

**Keywords:** attitudes, compliance and safety standards, copper mine, knowledge, respirable dust, work behavioral practices

## Abstract

Work in the copper mining industry is often associated with exposure to respirable dust and respirable crystalline silica. This exposure has the potential to cause silicosis, an incurable occupational respiratory lung disease. This study aimed at establishing the relationship between knowledge, attitudes, work behavioral practices and compliance with safety standards and workers’ exposure to respirable dust. A cross-sectional descriptive survey was conducted on 528 mine workers. The Statistical Package for Social Sciences (SPSS) was used for the descriptives. Structural equation modeling (SEM) with latent variables and partial least squares (LVPLS) analysis were employed to determine the relationship among these variables. The results indicated that of the four hypotheses, two were supported, and two were rejected, showing that there is a significant relationship between exposure to respirable dust and work behavioral practices, as well as compliance with safety standards. Knowledge and attitudes toward respirable dust exposure did not significantly influence exposure. According to the results from the survey, positive work behavioral practices as well as compliance with safety standards were significantly associated with exposure to respirable dust. It is recommended that mines should focus on the miners’ work behavioral practices and compliance with safety standards.

## 1. Introduction

Mining is considered one of the world’s most predisposed environments for diseases and injuries. The mining industry has recorded a significant proportion of nonfatal and fatal injuries worldwide. Whether workers are familiar with occupational health and safety (OHS) concepts or not, many organizations globally continue to record losses and inefficiencies due to non-compliance [1]. Exposure to crystalline silica dust exacerbates the risk of developing silicosis and cancer. Thus, the International Agency for Research on Cancer (IARC) has classified crystalline silica as a Group 1 lung cancer carcinogen [2].

To this effect, significant efforts are being made by the International Labour Organization (ILO) and World Health Organization (WHO) to prevent silicosis, an occupational respiratory disease which affects thousands of workers yearly. In 2020, a report by the ILO showed that annually, approximately 340 million occupational accidents occur, and 160 million victims succumb to work-related illnesses [3]. In 2019, the South African mining and quarry industries reported 2406 work-related injuries and 51 deaths in the mining industry due to accidents and occupational exposure to hazards [4]. In Zambia, the Occupational Health and Safety Act No 36 of 2010 in the Zambian Constitution covers welfare in terms of health and safety in the workplace [5], and Act No 271 covers compensation for occupation-related injuries and disease [6]. Two studies focusing on respirable dust and respirable crystalline silica quantification have been conducted in the Zambian context. The studies revealed overexposure to respirable dust and respirable crystalline silica [7,8]. Another study conducted in Zambia reviewed the medical data of 357 in-service miners by calculating a dose response risk for pulmonary tuberculosis (PTB) among the miners. It was observed that there was a significant increase with respirable dust exposure [9]. The legislation lacks policies and regulations on specific hazard exposures such as respirable dust and crystalline silica, which are capable of causing occupational respiratory diseases such as silicosis. Therefore, this is a gap that needs to be addressed.

Silicosis is considered one of the diseases miners can contract over time during mining operations. Silicosis is a preventable, incurable occupational respiratory disease caused by inhaling dust particles containing free crystalline silica from the environment [3]. Some environments require that workers are educated about exposure to respirable dust. The rationale behind the educational program is that the workers’ knowledge gained is perceived to assist them with changing “attitudes” and “work behavior” as well as seriously complying with health standards and regulations. Workers in developing countries are exposed to silica dust due to poor working conditions in industries such as mining and construction. Nguyen [10] stressed that many of the silica-exposed workers in the mining industry are unaware of the human health risks due to dust exposure.

The United Nations Sustainable Development Goal 3 Target 9 states that “by the year 2030 substantially reduce the number of deaths and illnesses from hazardous chemicals and air, water and soil pollution and contamination” [11]. The present study investigates the relationship between mine workers’ knowledge, attitudes, work behavior practices and compliance with safety standards based on respirable dust exposure. In the literature, miners’ vulnerability and cause of death due to respirable dust exposure have been well reported [12,13]. However, scientific evidence is scarce in understanding the knowledge, attitudes and behavior of employees regarding exposure to respirable dust [14]. Thus, this study aims to empirically establish if knowledge, attitudes and workplace behavior significantly impact how workers respond to health and safety propositions with regard to respirable dust exposure.

Exposure to respirable dust is a major challenge in the mining industry because it is a component of raw ore material. Crystalline silica is one of the components most widely encountered in minerals of the earth, with wide exposure in working environments and ambient environments [14]. In the past 15 years (i.e., 2008–2023), a significant number of studies on respirable dust exposure have been conducted in the mining and construction industries in some developing countries (e.g., South Africa, Ghana and Tanzania) and developed economies (e.g., China, Australia and the United States of America) [11,13]. According to the World Bank [15], the mining industry provides a significant contribution to the economic wealth of many countries. This significant contribution to mining economies has often been at the cost of the health of employees involved in the mining industry. The most common hazard is the inhalation of dust, which can lead to lung damage. Dust control measures are used to reduce dust exposure, and the effectiveness of these methods is dependent on how well they are used by workers, which is affected by their knowledge, attitudes and behavior.

A study conducted in Vietnam focused on workers’ knowledge, attitudes and practices in relation to silicosis among high-risk workers [10]. The outcome of the study was that knowledge, attitudes and practices were associated with high levels of health education, longer duration of work and not smoking. Furthermore, the study concluded that health education should be prioritized for workers. In a similar study, Ahadzi [16] conducted research on stone quarry workers’ awareness of the adverse effects of silica dust exposure. The research indicated that workers with secondary and higher education levels were more likely to be aware of silica dust exposure’s effects compared with workers without education.

In South Africa, Mavhunga [17] conducted a study on the knowledge, attitudes and practices of coal miners in relation to occupational health and safety at a Mpumalanga mining site. It was found that the miners demonstrated a lack of knowledge on health and safety legislation. Thus, they did not portray sufficient knowledge of occupational diseases which affect their attitudes and non-compliance. In addition, a study focusing on unsafe behavior propagation in miners’ social networks was conducted in China [18]. The outcome of the study was that knowledge played a mediating role in influencing unsafe behavior among the workers. In addition, a study by Takemura [19] examined the effects of wearing masks and worker education on preventing occupational dust exposure. The outputs of the study showed that workers need to be educated on proper usage of masks which may prevent worsening of pulmonary function due to dust exposure.

In the United States of America, a study on formative research to reduce mine workers’ respirable silica dust exposure was conducted by Haas [20]. The research focused on assessing the feasibility of integrating technology into behavior interventions, and one of the objectives was to determine mine workers’ knowledge and attitudes toward respirable silica dust exposure. The participants were found to be knowledgeable on activities that may expose them to respirable silica dust, and workers complied with safety standards toward dust exposure by wearing respirators to a point that they wore respirators even when it was not required. It was also revealed that the workers who had served longer years showed increased positive attitudes toward personal health compared with those that had fewer service years.

In another study focusing on mining engineering students being prepared for the mining sector, the students’ knowledge, attitudes and practices regarding occupational safety and health issues were investigated [21]. The authors of the research showed that field work experience, interest in occupational safety and health and webinar attendance contributed to improving occupational safety and health knowledge, attitudes and practices in the mining industry.

The study by Sunandar and Yodang [21] used a limited sample size of 56, making deriving generalizations from the findings difficult. Two studies by Mavhunga and You [17,18] used cross-sectional research designs, which limited the conclusions for any cause-and-effect relationships between the variables (e.g., knowledge and unsafe behavior) [19]. Lastly, the study by Haas [20] was limited to respondents with knowledge of technology because the questionnaire was administered online, excluding respondents who were unable to use the online platforms but could give relevant information on the topic. Compliance with safety standards is an equally important aspect in OHS to knowledge, attitudes and work behavioral practices because it helps to prevent accidents in a workplace. From the studies reviewed, no study focused on how compliant mine workers are in relation to hazard exposure. This study, therefore, incorporated compliance with safety standards as one of the moderators in reducing worker susceptibility to respirable dust exposure. This study also used a self-administered questionnaire with the help of knowledgeable research assistants present to help answer any questions that may have seemed unclear in the questionnaire for some respondents. 

### 1.1. Theoretical Background

The theory provides an essential roadmap for investigating elements that can either prevent or promote the safety and health of workers in a workplace. This section presents different theoretical models and frameworks that have been used in similar studies. The first part discusses three used theoretical models and frameworks, which include the Health Belief Model (HBM), the Theory of Planned Behavior (TPB) and the Transtheoretical Model/Stages of Change Mode (TTM/SOC). The second part discusses the conceptual model used in the study.

#### 1.1.1. Health Belief Model (HBM)

The Health Belief Model (HBM), opined in the 1950s by Irwin M. Rosenstock, Godfrey M. Hochbaum, S. Stephen Kegeles and Howard Leventhal, is widely used in determining the health behavior of a person and postulates that the health behavior of individuals is the consequence of perceptions, modifying behavior and the likelihood of action [21]. The HBM focuses on four different dimensions, which include (1) perceived susceptibility of how one individual’s vulnerability to contracting a disease differs from the others, (2) how the perceived severity of contracting a particular disease for an individual may differ from one person to another, which also applies to (3) perceived benefits of acceptance of susceptibility toward a particular condition, and (4) beliefs regarding the effectiveness of a particular action also differing between individuals and perceived barriers, which is potential negativity toward undertaking a particularly recommended behavior [22]. In OHS, the HBM helps with explaining the safety behavior of workers. The HBM was selected as one of the underlying theories for the study because of its emphasis on perceived susceptibility as one of the influencers of behavior in a workplace. This model helps to view a situation according to personal perception. The relevance of this model with the study is that miners’ behavior toward respirable dust exposure plays a pivotal role in their health, and individual workers are presumed to experience difficulties with breathing in a work environment where airborne particles are constantly present. Thus, all four dimensions highlighted above are likely to prevail. In such a situation, the pollution levels presumably will be high, hence making it riskier for the workers.

#### 1.1.2. Theory of Planned Behavior (TPB)

The Theory of Planned Behavior (TPB) was postulated by Icek Ajzen, and it helps to understand behavior changes in humans. This theory helps provide guidelines which help to examine the beliefs and behavior of individuals [23]. It postulates that attitudes toward a certain behavior, subjective norms and perceived behavior guide a person to act in a particular way [24]. Thus, the purpose of displaying an action is determined by attitudes toward one’s action, subjective norms, which are a person’s social influences or beliefs, and perceived behavior control, which is how easy or difficult it is for one to engage in a behavior. This theory highlights that for a person to behave in a particular way, attitudes, subjective norms and perceived behavior control must be combined. It is a fact that mining companies are equipped with human capital, which is a very important resource and calls for safe behavior in a workplace environment. Therefore, applying the TPB helps to determine the level of workplace safety and understanding of human behavior changes, and in this study, it will be used with regard to the behavior of miners toward respirable dust. Like the Health Belief Model, the TPB is also perceived to be a relevant theoretical framework that directs the factors to be considered in the conceptualization of the research model to guide the hypothesis formulation.

#### 1.1.3. Transtheoretical Model/Stages of Change Model (TTM/SOC)

The Transtheoretical Model/Stages of Change Model (TTM/SOC) was developed by James Prochaska and his colleagues. It theorizes that temporal behavior integrated with processes and principles can change an individual’s behavior. According to this theory, people are at different stages in acquiring a particular behavior [25]. Behavior change is deemed to be a process that combines several nonlinear steps over a long period. The TTM can be explained using six stages: (1) pre-contemplation, a stage that involves one’s aim of not taking action in the next 6 months, (2) contemplation, which involves the probability or chance of taking action in 6 months, (3) preparation, which is organizing for change in the next month, (4) action, which is taking on new behaviors in less than 6 months, (5) maintenance, which adopts a change in health behavior and takes it up for more than 6 months, and (6) termination, which involves continuing with a behavior achieved over time [26]. The six processes of change are capable of associating a person’s progression from one phase to another. In this study, the TTM will be used to determine the health behavior of miners in relation to their knowledge of and attitudes and compliance toward respirable dust.

#### 1.1.4. Model Used in the Study

Based on the three theories, some investigators proposed that unifying the models may provide a sure understanding of predicting behavior in a workplace [26,27]. Dejoy [28] extracted five major constructs from the HBM, TPB, and TTM as well as the Theory of Reasoned Action (TRA), which were combined with the four stages of workplace self-protective behavior ((1) the hazard appraisal stage, (2) the decision-making stage, (3) initiation of self-protective action and (4) adherence of self-protection, adopted from the TTM) and created an integrative health protective behavior model [28]. Five constructs from various theoretical models are combined in Dejoy’s integrative model. These include threat-related beliefs about the severity of and vulnerability to hazards, response effectiveness for how a person views the benefits of self-protective behavior, facilitating conditions for expanding concepts to be used in value expectancy models and the safety climate, which is a combination of social and organizational factors [29]. Based on the guidelines from Kim [27] and Dejoy [28], this study pursued a similar approach to integrate some of the theory tenets from the three theories. To this effect, this study reviewed the miners’ knowledge, work behavior, attitudes and compliance toward the environment both at the surface and underground, as characterized by respirable dust. All mine workers have a significant role to play in a particular behavior, which results in a blend of different behavior factors. Figure 1 illustrates a constitution of the main thrusts, such as behavior, attitudes and compliance in the three models (i.e., TPB, HBM and TTM). These models were used because miners’ knowledge, behavior, attitudes and compliance toward respirable dust exposure play a vital role in their health. Based on the factors explaining respirable dust exposure (knowledge, attitudes, workplace behavioral practices and compliance with safety standards), a conceptual model was formulated. Moderator factors were linked to helping explain the relationship that exists between exposure to respirable dust and knowledge. Workplace behavior has been linked to occupational health and safety by some studies.

The conceptual framework stemming from the theoretical framework was based on factors that can help determine the respirable dust exposure based on attitudes, work behavioral practices and compliance regarding respirable dust exposure, which were taken as the primary predictors of a worker’s intention to engage in a particular behavior [30]. An outcome of a certain behavior in a workplace can either be positive or negative, and therefore linking three predictors to respirable dust exposure can be one of the major contributors to either being susceptible to respirable dust exposure or not [31]. Based on the concepts, this study measured workers’ emotional and environmental responses to stimuli (i.e., the effects of respirable dust). Definitions of the key terms used in the conceptual model in context with this study are given in Figure 1.

##### Attitudes

Attitude has been defined by different publications, such as those by Ajzen and Fishbein [32] and Hogg and Vaughan [33]. According to Ajzen and Fishbein [32], attitude refers to a learned tendency that makes an individual respond in a consistently favorable or unfavorable manner concerning a given object. Hogg and Vaughan [33] defined attitude in a broader sense as a “mental and neutral state of readiness put up through experience and exerting an active or directive influence upon the individual’s reaction in a situation” In addition, attitude has been defined as a relatively enduring organization of norms, beliefs and behavioral tendencies toward socially significant situations, groups, events or symbols [34]. The definition of attitude from a psychological perspective is a set of emotions, beliefs and behavior toward a circumstance [35]. Given the different definitions, this study adopted the psychological perspective of attitude as forces arising from internal emotions and beliefs described by Cherry [35]. This study focuses on how miners’ attitudes affect their respirable dust exposure.

##### Behavior

The term behavior has become a commonplace feature in public health studies, and its definitions vary. In several definitions, the target focus has been the movement of an individual due to stimuli. Nonetheless, efforts were made to identify and review some definitions to give the contextual definition for this study. Some definitions include that of Tinbergen [36], who stated that behavior is a total movement made by an animal in its entirety. Contrary to this, Alder [37] highlighted that Dretske (1988) defined behavior as a process of an inner entity bringing about bodily movement or an environmental outcome. Furthermore, Shull [38] stated that behavior is an activity that an individual engages in. This study adopted the psychology perspective of behavior from Drestske, which states that “behaviour comprises of an organism’s external reaction to its environment”. This study, therefore, operationalized behavior as how miners interact with their dusty environment.

##### Compliance

Cialdini and Goldstein [39] defined compliance as “a kind of response to a particular communication upon request which may give an explicit or implicit outcome”. Interestingly, for the definition of psychology, Cherry [35] disclosed that compliance is changing one’s behavior at the request or instruction of another person. Psychologists perceive compliance as a change in behavior because another person asked him or her to do so. Based on the interrelatedness of the definition, this study adopted the simple definition propagated by the field of psychology, which states the following: “In psychology, compliance refers to changing one’s behavior at the request or direction of another person”. Miners’ compliance with safety behavior is cardinal to their safety. Therefore, this depends on their knowledge of health risks, attitudes and behavior, as indiated by Cherry [35].

Given the aforesaid information, a safe environment in a workplace is presumed to greatly influence how a worker will behave toward safety behavior. Considering the three behavior models adopted and modified (HBM, TPB and TTM), they can be helpful for foreseeing and understanding factors likely to positively improve the health of workers in a workplace. This justifies why modified HBM, TPB and TTM models formed the study framework in addressing the occupational, health and safety challenges of respirable dust exposure. It is in this vein that the following hypotheses emerged.


*Knowledge of risks and dangers of respirable dust and respirable dust exposure by mine workers*


The concept of being knowledgeable about potential hazards in occupational health and safety in a workplace, as stated by Aluko, is cardinal to having a positive attitude that may help workers portray a particular behavior [40]. According to Nguyeni [10], a hazard or disease can be controlled if workers are knowledgeable about the risks and dangers of a particular hazard. To this effect, Takemura [19] carried out a study on factory workers’ occupational dust exposure. The study concluded that interventions for educating the workers helped with the prevention of respiratory diseases among the workers. Based on the aforementioned explanations highlighted by Aluko [40], Nguyeni [10] and Takemura [19], the following hypothesis was postulated:

**H_1_:** *Knowledge of the risks and dangers of respirable dust positively reduces exposure to respirable dust in workers*.


*Knowledge, attitudes of workers and respirable dust exposure for mine workers*


The basis of attitudes in the framework shows that workers with positive attitudes are more likely to be cautious with the way they handle a hazard or disease. In addition, it is perceived that attitudes change due to the knowledge that the worker is provided within a workplace about the hazards present and the opportunity to be self-conscious and aware of dust-prone work environments. Hence, the wearing of personal protective gadgets like respirators is always upheld by the worker [10]. People assess outcomes and determine how they will perform in a particular situation. Thus, a person who believes in performing a particular behavior will have positive outcomes, whereas an individual believing that performing a particular behavior result in negative outcomes will hold a hostile attitude toward the behavior. The reasoning behind this approach is that one’s attitude can greatly influence the decisions they make in a work environment. Based on this discussion, we hypothesized the following:

**H_2_:** *Knowledge of the risks and dangers of respirable dust is positively moderated by attitudes to reduce respirable dust exposure in workers*.


*Knowledge, work behavioral practices and respirable dust exposure for mine workers.*


The concept of safety behavior has been put forward as the way someone acts to support or not support safety practices (e.g., in training for safety or in complying with safety activities). Safety practices are key to minimizing workplace injuries and accidents. Work behavior contributes to accidents in a workplace. Therefore, the presence of good safety behavioral practices results in workers complying with safety rules [41]. Zin ad Ismail [41] further stated that employees need a particular behavior to engage in good safety practices. Hence, we hypothesized the following:

**H_3_:** *Knowledge of the risks and dangers of respirable dust is positively moderated by work behavioral practices toward respirable dust exposure for workers*.


*Knowledge, compliance with safety standards and respirable dust exposure for mine workers*


Safety compliance is the act of sticking or adhering to safety rules set by an overseeing body or legislators to protect the welfare of workers in a workplace. Companies need to adhere to standards set by regulatory bodies for them to be considered compliant, and this has to be carried out with the help of safety officers enforcing the regulations set. Workers tend to engage in a belief that an accident cannot happen to them, which justifies them adopting unsafe behavior in workplace practices. Hence, the following hypothesis was postulated:

**H_4_:** *Knowledge of the risks and dangers of respirable dust is positively moderated by compliance with safety standards toward respirable dust exposure*.

## 2. Methods

The methodology used during this study was a cross-sectional survey of mine workers in both underground and surface operations at a mining site located in the Copperbelt Province of Zambia. Zambia is a landlocked country located in the southern region of Africa. The population included all miners working in dusty jobs who were purposely chosen. The sample frame focused on miners working in highly dusty environments such as operators, rock breakers, conveyer belt attendants, workmen, the person in charge, section bosses, shift bosses, mine captains and engineers. The total target for respondents was 600, which was slightly over half of the population, to include a representative sample of the total population of 1041 miners in dusty jobs. Thirty-eight qualitative statements based on the hypotheses from the conceptual framework were grouped under the knowledge of the risks and dangers of respirable dust, attitudes toward working in a respirable dust environment, working behavioral practices in the workplace, compliance with safety standards and exposure to respirable dust. The statements were measured using a 5-point Likert scale questionnaire, with the lowest value of 1 representing “strongly disagree” and the highest value of 5 representing “strongly agree”, and this is provided in the Appendix A.

The questionnaire was first piloted on 50 mine workers, with 25 surface and 25 underground workers from one mining shaft. The respondents highlighted that some questions needed clarity, but the time to complete the questionnaire was good because it was completed within 7 min, which did not interfere with their work schedules. Compliance as a topic was missing in the questionnaire, and thus it was added later. Six hundred potential respondents were given the revised questionnaire through the human resource department. The actual responses received were 528 out of the 600 questionnaires handed out, and thus the response rate was 88%. The respondents answered all the questions, indicating that there were no missing values. The respondents included 217 (41%) surface mine workers and 311 (59%) underground mine workers.

## 3. Data Analysis and Results

The collected data were entered into Microsoft Excel and later exported to Statistical Package for the Social Sciences (SPSS) for descriptive analysis and structural equation modeling (SEM), with the latent variables in partial least squares (LVPLS) embedded in SmartPLS4 for model analysis. The data were initially tested for the presence of outliers using skewness and kurtosis. This test proved that the data were normally distributed. Secondly, the data were subjected to validity and reliability tests. This helped reduce the variables from the initial 38 using a cutoff factor loading of 0.7. Three factors—K1 (K = knowledge), ERD1 exposure to respirable dust) and ERD8—were deleted because this study aimed at exploring relationships at a high factor loading.

Therefore, the average factor loading was specified at the 0.7 threshold. BP4 was eliminated because it was a multi-collinear statement with a variance inflation factor (VIF) greater than five. Thirty-six factors were loaded as summarized in Table 1. Loading relevance was conducted on the 35 indicator factors to check if the model could be maintained using PLS model 4. Indicators with problems were identified and removed from the PLS to improve the value of the average variance extracted (AVE) and composite the reliability of their constructs over the 0.5 thresholds as shown in Figure 2.

Additionally, the internal consistency of the model to determine the composite reliability for the constructs of knowledge, attitudes, behavioral practices, compliance safety standards and exposure to respirable dust were estimated to be 0.942, 0.932, 0.919, 0.936 and 0.893, respectively. From the results, it was observed that the internal consistency reliabilities for the factors were very good.

### 3.1. Descriptive Analysis

The sample was characterized by 217 surface and 311 underground miners, with 431 male and 97 female mine workers, where 383 were permanently employed while 145 were on temporary employment. In addition, 37 attained a primary education level, 271 attained a secondary education level, and 220 attained up to a tertiary education level. The mean scores for the responses to each questionnaire item and the overall score for each variable were computed. According to Likert [42] and Boone and Boone [43], Likert scale items can be computed and interpreted as follows: 1–1.8 (strongly disagree), 1.9–2.6 (disagree), 2.7–3.4 (neutral), 3.5–4.2 (agree) and 4.3–5 (strongly agree).

It can be seen that the majority of the workers agreed with the statements on the questionnaire about knowledge, attitudes, work behavior and compliance with safety standards, since the mean scores of the entire group ranged between 3.5 and 4.2. Concerning respirable dust exposure, the majority of the workers gave neutral responses. It was observed that the mean scores for the underground workers were slightly higher for all variables compared with the surface workers.

### 3.2. Outer Loadings

The outer loadings for knowledge ranged between 0.812 and 0.898, attitudes ranged between 0.772 and 0.923, behavioral practices ranged between 0.721 and 0.866, compliance and safety standards ranged between 0.769 and 0.882, and reduced exposure to respirable dust ranged between 0.702 and 0.867.

### 3.3. Convergent Validity

Convergent validity was determined using average variance extracted (AVE). If the AVE for the latent variable is greater than the 0.5 threshold, then the variable can explain the variability. The following AVE values were obtained: knowledge was 0.699, attitudes reached 0.692, behavioral practices reached 0.665, compliance and safety standards reached 0.633, and exposure to respirable dust was 0.530. Therefore, the measures of the five variables showed that the constructs recorded high levels of convergent validity because they were all above the 0.5 threshold.

### 3.4. Discriminant Validity

This study went further to assess the relationship between the indicator variables, which is a requirement in PLS-SEM. This was achieved by applying the Fornell–Larcker criterion to assess discriminant validity, as shown in Table 2. Discriminant validity was met because the square root of the AVE for the indicator variables knowledge, attitudes, behavioral practices, compliance and safety standards and exposure to respirable dust were larger than the corresponding latent variable correlations (LVCs).

### 3.5. Collinearity Assessment for the Structural Model in PLS-SEM

Collinearity was tested using the variance inflation factor (VIF). This is a significant test in the structural model where if the VIF value is above five, then a factor is considered to be collinear with other factors and therefore problematic. The collinearity results are summarized in Table 3. From the Table, one can see that the VIF values were less than five, suggesting that attitudes, behavioral practices, compliance and safety standards and knowledge were not collinear. All the values were less than five for each variable.

### 3.6. Coefficient of Determination (R^2^)

In the model in PLS-SEM, exposure to respirable dust was the dependent variable. How much change was accounted for by each predictor variable was explained using the coefficient of determination (*R*^2^). The rule of thumb for *R*^2^ is that the values 0.25, 0.50 and 0.75 are considered to be weak, moderate and strong coefficients of determination, respectively. In this research, an *R*^2^ value of 0.187 was considered weak for knowledge, attitudes, behavioral practices and compliance and safety standards. This means that by considering the variables jointly, they explained 18.7% of the variance of the dependent variable exposure to respirable dust. In other words, there is a likelihood that other factors not considered as variables for this study might have accounted for the remaining 81.3%.

### 3.7. Path Coefficient

From the conceptual model, the results indicate that two out of four structural models had significant relationships, such as between behavioral practices and exposure to respirable dust, compliance and safety standards, knowledge and attitudes, knowledge and compliance and safety standards and knowledge and behavioral practices. Attitudes and exposure to respirable dust and knowledge and exposure to respirable dust were non-significant, as summarized in Table 4.

### 3.8. Predictive Relevance (Q^2^)

The predictive relevance *(Q*^2^*)* establishes the predictive power of the endogenous constructs in a model. If Q^2^ is greater than zero, then this indicates that the values in the model were well constructed and that the model has predictive power. Attitudes (0.682), behavioral practices (0.520), compliance and safety standards (0.470) and exposure to respirable dust (0.100) were the only endogenous constructs that were used in the model for running PLS-predict. They all indicated good predictive relevance for the endogenous variables because the Q^2^ values were all greater than zero.

### 3.9. Effect Size

The effect size determines how variables influence each other in a model. The effect size can be interpreted as 0.02, 0.15 and 0.35, which are small, medium and large effects, respectively. The structural model was explored to assess the effect of how exogenous constructs affected endogenous constructs. It was observed that knowledge had a large effect (3.514) on attitudes, behavioral practices and compliance and safety standards (1.528 and 1.152), while a small effect size was observed on attitudes (0.004), behavioral practices (0.013), compliance and safety standards (0.004) and knowledge (0.004) regarding exposure to respirable dust.

### 3.10. Multi-Group Analysis

This data set comprised mine workers in dusty jobs for both underground and surface operations. Therefore, it was imperative to conduct multi-group analysis (PLS-MGA) using a parametric approach. This approach compares the path coefficients of different groups by modifying an independent-sample *t* test using bootstrapping. The standard deviation of the path coefficients was calculated to see if the surface and underground, male and female or education level factors had moderating effects on the findings of the research. Concerns arose because heterogeneity may have existed in modeling the significant differences of the categorical groups. The results revealed that in terms of location of operations, the underground miners and surface miners differed in knowledge of the risks and dangers of respirable dust exposure (*p* < 0.05) and exposure to respirable dust (*p* < 0.05). In relation to gender, there were no significant differences, and in relation to education level, significant differences were seen between those with secondary and tertiary education in relation to attitudes and compliance and safety standards (*p* < 0.05).

### 3.11. Hypotheses Testing Results

Out of the four main hypotheses generated in the conceptual model, two were supported and two were not supported, as shown in Table 5. Knowledge was found to have significant relationships with attitudes, behavioral practices and compliance and safety standards. Therefore, the hypothesis was retained. Nonetheless, the relationships between knowledge and attitudes toward exposure to respirable dust were non-significant, and hence, the hypothesis was rejected.

## 4. Discussion

Concerning the descriptive statistics given in Table 1, the majority of the workers knew the health risks associated with respirable dust exposure. They had positive attitudes towards wearing respiratory protective equipment (RPE), and the company provided RPE whenever necessary. For work behavior, most of the workers showed good behavior and practices for the dusty environments where they worked, and the majority of the workers complied with the safety standards of dust exposure. It is important to note that these workers showed a neutral stance toward the aspect of exposure to respirable dust items because they were not sure if they were exposed to dust the entire shift or not or if they needed job rotations during operations. There were no significant differences in gender, age or education level. However, the groups (surface and underground miners) differed in the mean values about knowledge, work behavioral practices, compliance with safety standards and work area. Work area was one of the variables that was explored, where the mean scores for the underground miners were, on average, higher than those for the surface miners. This implies that the underground miners were more knowledgeable of the risks related to respirable dust exposure than the surface miners. They had better attitudes toward wearing PPE than surface miners, the work behavior and practices toward exposure to respirable dust were better than those of the surface miners, and they were more compliant with safety standards than the surface miners.

**H_1_:** *Knowledge of the risks and dangers of respirable dust positively reduces exposure to respirable dust in workers*.

The results for the first hypothesis show that there was no significant relationship between knowledge of the risks and dangers of respirable dust and exposure for the miner workers studied. The results show that workers may have knowledge of the risks and dangers of respirable dust exposure, but it may not have an impact on their exposure to respirable dust during operations. However, the findings of this study are contradicted by Ahadzi [16], who established a relationship between workers’ awareness of silica dust and workers’ knowledge of how it adversely affects their geographical location in two Ghanaian regions. Another study that contradicts the findings is that of Haas [20], which indicated a relationship between knowledge of respirable silica dust and sources of exposure among workers. Even though the hypothesis was not supported in this study, similar studies have shown that the management in work places seriously undertake appropriate measures for educating workers on the risks and dangers of respirable dust exposure because this can reduce the chances of contracting respiratory diseases.

**H_2_:** *Knowledge of the risks and dangers of respirable dust is positively moderated by attitudes to reduce respirable dust exposure in workers*.

Knowledge is important to creating positive attitudes that will infuse good behavior in a workplace and reduce respirable dust exposure. The results in Table 5 show that knowledge was not positively moderated by attitudes to reduce respirable dust effects on miners, implying that even though the workers know, their attitudes will not necessarily influence exposure to respirable dust. The findings of the current study are in contrast with what Mavhunga [17] found in a coal mine, where the workers had a good appreciation of attitudes toward occupational exposure.

**H_3_:** *Knowledge of the risks and dangers of respirable dust is positively moderated by work behavioral practice towards respirable dust exposure*.

Hypothesis 3 was supported, stating that the workers’ knowledge was moderated by positive perceived behavior toward respirable dust exposure, as seen in Table 5. This is supported by the findings of Haas [20], who found that mine workers were knowledgeable about health-protective behavior toward respirable silica dust exposure to the point that they ensured that they wore their respirators even in areas where it was not required. Furthermore, the study conducted by You [18] in China disclosed that knowledge of safety played a role as a mediator in the unsafe behavior propagation of mine workers. This result means that the management of mining sites should continue with awareness programs on the knowledge of the risks and dangers of respirable dust exposure so that miners can integrate them into positive behavior and practices. Table 5 shows that there was a significant relationship between the variables. This result is in agreement with the finding of the study by You [18], which focused on miners’ unsafe behavior regarding safety during operations at a mining site in China. The authors found that unsafe behavior propagation significantly affected the safety of miners. The outcome of this hypothesis implies that the perceived behavioral practices of miners should remain positive toward reducing respirable dust exposure during mining operations.

**H_4_:** *Knowledge of the risks and dangers of respirable dust is positively moderated by compliance with safety standards toward respirable dust exposure*.

Table 5 shows that the knowledge of miners regarding the risks and dangers of respirable dust is positively moderated by compliance. Therefore, concerning compliance with safety standards, workers at the mining site need to comply with the regulations on respirable dust to reduce exposure. This finding is supported by the study conducted in Siaya County in Kenya, which found that knowledge was one of the key indicators that influenced regulatory compliance regarding the safety of miners [44]. This is consistent with our study regarding knowledge and compliance with safety standards. The outcome of the hypothesis implies that miners need to know the health risks to comply with the safety standards at the mining site. Beth [44] recommended that the predictors influencing compliance with safety regulations in small-scale mines were concerned with the perceived cost of compliance, awareness of safety requirements and administrative failures, which aligns with the current study. Since the result was significant, there is a need for mining companies to make compliance with safety standards regarding exposure to respirable dust a priority during mining operations to reduce exposure to respirable dust.

The current study aimed at empirically establishing the relationship between knowledge, attitudes, work behavioral practices and compliance with safety standards for respirable dust exposure at mining sites. The findings of this study revealed that there was no direct significant relationship between knowledge of the risks and dangers of respirable dust, the attitudes of workers toward respirable dust and exposure to respirable dust. Hypotheses one and two contradicted the findings of Ahadzi [16] and Mavhunga [17]. This could have been led by contextual or methodological differences and differences in the use of variables. However, there were significant relationships between working behavioral practices at the workplace and compliance with safety standards and respirable dust exposure of workers. This implies that workers have knowledge of the health risks associated with respirable dust exposure through awareness programs at the mine, but there is a need to continuously monitor the work behavior of miners concerning compliance with standards such as the routine wearing of PPE during operations.

Another observation from the study was that work behavioral practices and compliance with safety standards were moderators between knowledge of the risks and dangers of respirable dust exposure and reduced susceptibility to respirable dust. Therefore, being knowledgeable of the risks and dangers of respirable dust exposure is not enough to reduce exposure among workers, but being knowledgeable of promoting good work behavioral practices and implementing compliance with safety standards will reduce worker exposure to respirable dust because the knowledge of ownership is not good enough to influence exposure. Therefore, mining companies should embark on prompt awareness monitoring programs concerning knowledge on good worker behavioral practices which will highlight how workers should behave during operations and give them the consequences for how they will deal with workers not complying with safety standards.

This study recommends that the management of a mining company, through awareness programs, should ensure that workers’ attitudes and work behavioral practices are monitored constantly so that respirable dust exposure is reduced to reasonable standards during operations. This can be achieved with standardized continued monitoring programs. On this basis, future studies should be conducted in different mining companies across Zambia to determine how these factors influence respirable dust exposure in other mines. This study gives an update of the occupational safety and health status in Zambia and helps employers develop health models for their employees. In addition to this, future studies should include measurement data to relate to the findings of this study and observations, interviews and focused group discussions. Furthermore, other studies should explore the current conceptual model by determining respirable dust’s effects on mine workers’ compliance and safety standards through knowledge and behavioral practices.

### Limitations

The limitation of this study is that it measured knowledge, attitudes, work behavioral practices and compliance from miners at one mining site, which makes it difficult to generalize results to other mining sites.

## Figures and Tables

**Figure 1 ijerph-20-06785-f001:**
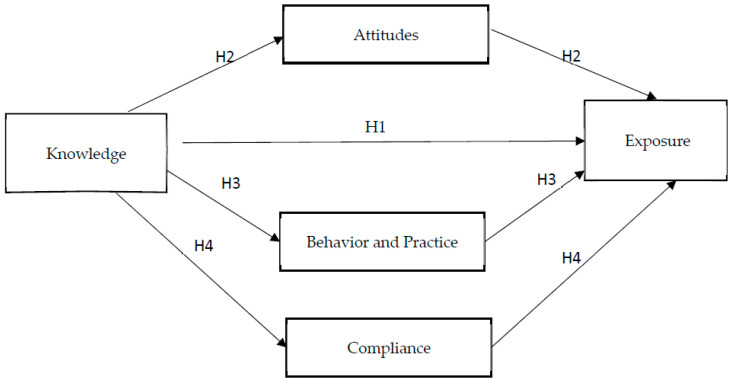
The relationship between knowledge and exposure to respirable dust exposure among mine workers, which is moderated by attitudes, behavior, practices and compliance.

**Figure 2 ijerph-20-06785-f002:**
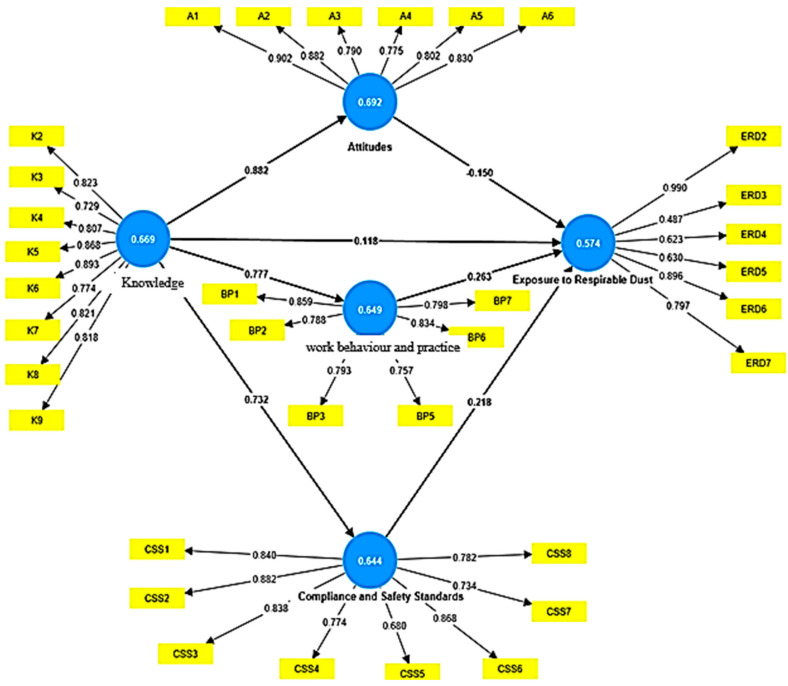
PLS path model estimation.

**Table 1 ijerph-20-06785-t001:** Mean scores for individual items and overall variables organized by work area.

Variable	Work Area	Mean for Individual Items (Range)	Overall Mean Score	Overall Standard Deviation
Knowledge	All	3.31–3.59	3.50	1.14
	Underground	3.31–3.59	3.58	1.13
	Surface	3.43–4.49	3.39	1.15
Attitudes	All	3.37–3.80	3.71	1.15
	Underground	3.72–3.80	3.76	1.01
	Surface	3.37–3.70	3.60	1.15
Work behavioral practices	All	3.29–3.64	3.50	1.16
	Underground	3.49–3.64	3.59	1.20
	Surface	3.29–3.41	3.38	1.09
Compliance with safety standards	All	3.41–3.64	3.53	1.06
	Underground	3.31–3.64	3.57	1.09
	Surface	3.41–3.49	3.47	1.03
Exposure to respirable dust	All	3.20–3.57	3.38	0.96
	Underground	3.45–3.57	3.48	0.92
	Surface	3.20–3.42	3.22	0.99

**Table 2 ijerph-20-06785-t002:** Fornell–Larker criterion.

	Attitudes	Behavioral Practices	Compliance and Safety Standards	Exposure to Respirable Dust	Knowledgeability
Attitudes	0.865				
Work Behavior and Practices	0.778	0.845			
Compliance and Safety Standards	0.751	0.835	0.831		
Exposure to Respirable Dust	0.325	0.391	0.391	0.806	
Knowledge	0.828	0.724	0.688	0.326	0.843

**Table 3 ijerph-20-06785-t003:** Collinearity assessment.

	VIF
Attitudes	3.238
Work Behavioral Practices	2.839
Compliance and Safety Standards	2.849
Exposure to Respirable Dust	2.227
Knowledge	2.958

**Table 4 ijerph-20-06785-t004:** Path coefficients and significance testing results.

	Sample Mean (M)	Standard Deviation (STDEV)	t-Statistics (|O/STDEV|)	*p* Values
Knowledge -> Exposure to Respirable Dust	0.085	0.079	1.103	0.27
Knowledge -> Attitudes	0.828	0.015	54.139	0.00
Knowledge -> Work Behavioral Practices	0.723	0.026	27.627	0.00
Knowledge -> Compliance and Safety Standards	0.688	0.027	25.61	0.00
Attitudes -> Exposure to Respirable Dust	−0.058	0.079	0.751	0.453
Work Behavioral Practices -> Exposure to Respirable Dust	0.2	0.083	2.407	0.016
Compliance and Safety Standards -> Exposure to Respirable Dust	0.211	0.086	2.429	0.015

**Table 5 ijerph-20-06785-t005:** Results of hypotheses testing.

	Hypotheses	Remark
H_1_	Knowledge of the risks and dangers of respirable dust positively reduces exposure to respirable dust in workers.	Not Supported
H_2_	Knowledge of the risks and dangers of respirable dust is positively moderated by attitudes to reduce respirable dust exposure in workers.	Not Supported
H_3_	Knowledge of the risks and dangers of respirable dust is positively moderated by work behavioral practices for respirable dust exposure.	Supported
H_4_	Knowledge of the risks and dangers of respirable dust is positively moderated by compliance with safety standards for respirable dust exposure.	Supported

## Data Availability

Data are available upon reasonable request from the corresponding author due to ethical restrictions.

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
