# Peer review of "Knowledge, Attitude, Behavior Practices and Compliance of Workers Exposed to Respirable Dust in a Zambian Copper Mine"

_ijerph, 2023, doi:10.3390/ijerph20186785_

Round 1
Reviewer 1 Report
PEER REVIEWER 2 – Comments
1. I did not see any literature or studies that shows the outcome of interest discussed regarding the study setting Zambia. Are there no literature review conducted in Zambia as regards this public health concerns? Are there no medical studies or non-medical studies conducted in this areas….I doubt….kindly adjust the introduction…I think you should merge the literature review and the introduction together. The paper should have introduction, Methods, Results……..[Line 31-163, Page 1-4]
2. Authors should rewrite this subsection on the theories or models used in this study and how you used them. You were just explaining what others have used and some of the theories other authors have used were not mentioned in your work. Specifically, mentioned the type of theories(s) you used and explained then under the section ‘Theoretical framework/model used in this study’ [Line 220-254, Page 5-6]
3. Who propounded the Health Belief Model. What was the reasons the theory was propounded by the authors? How is it applicable in your study and how do you want to operationalize it in your study? [Line 173-174; Page 4]
4. Who propounded the Theory of Planned Behaviour. What was the reasons the theory was propounded by the authors? How is it applicable in your study and how do you want to operationalize it in your study? [Line 191-195; Page 4]
5. Who propounded the Transtheoretical Model/Stages of Change Model (TTM) What was the reasons the theory was propounded by the authors? How is it applicable in your study and how do you want to operationalize it in your study? [Line 207-211, Page 5]
6. You did not make use of Theory of Reasoned Action (TRA), so why discussing it on the section of model used in the study [Line 223-224, Page 5]
7. The conceptual framework show stem from the theoretical framework, we should see some of the concepts of the three theories you utilized in this your study. Address it. [Line 243-251, Page 5-6]
8. In Figure 1, these variables in the framework, where did you derive them from? Did they stem out of the three theories you used? It should stem from your theories….. [Page 6]
9. The concepts, attitudes, behaviour and compliance, I think what you did was conceptual definition or how other studies have utilized it in their studies. But my question to you is how do you want to apply it to your study? How did you conceptualised these variables from your theoretical models or frameworks and operationalise it in your study [Line 255-293; Page 6-7]
10. How did you measure knowledge, attitude and compliance in this study [Line 295-343; Page 7-8]
11. Describe the questionnaire employed in data collection? How did you derive at the questionnaire, did you get it from a primary, secondary or tertiary sources? How did you validate the questionnaire? [Line 357-366; Page 8]
12. Under Methods, I expect to see geographical description of Zambia, then followed the study design and population sampling, then followed by Sample size and inclusion/Exclusion criteria and Data collection, variable measurement and Data analysis as well as Ethical considerations. The methods should follow the scientific process….a process it follows from one stage to another…you cannot jump one stage and move to another stage… [Line 344-389, Page 8]
13. How did you arrive at 600 questionnaires? [Page 8]
14. Under results, Table I should be the social demographic characteristics of the respondents…let us know the characteristics of the study population [Page 9]
15. Arrange the results according to your objectives/Hypothesis so that it can be understandable
16. Check this Table 3.1, the first column…I think reduce the font size to occupy the many columns [Page 9]
17. The discussion section should be aligned with the present study findings with each objective discussed, separated by paragraph [Line 497]
18. Remove the entire sentences from the manuscript [line 542-543]
19. Where are the strengths of this study? [Line 609]
20. Check Reference 42
21. Check this https://www.preprints.org/manuscript/202307.0827/v1. Please indicate in your manuscript that this paper has been preserved in preprints
22. See Recommended Readings
1. Brouwer, D. H., & Rees, D. (2020). Can the South African Milestones for Reducing Exposure to Respirable Crystalline Silica and Silicosis be Achieved and Reliably Monitored? Frontiers in Public Health, 8, 526930. https://doi.org/10.3389/fpubh.2 020.00107
2. Entwistle, J.A., Hursthouse, A.S., Marinho Reis, P.A. et al. Metalliferous Mine Dust: Human Health Impacts and the Potential Determinants of Disease in Mining Communities. Curr Pollution Rep 5, 67–83 (2019). https://doi.org/10.1007/s40726-019-00108-5
3. Ngosa, K., Naidoo, R.N. The risk of pulmonary tuberculosis in underground copper miners in Zambia exposed to respirable silica: a cross-sectional study. BMC Public Health 16, 855 (2016). https://doi.org/10.1186/s12889-016-3547-2.
4. Hayumbu, P., Robins, T. G., & Key-Schwartz, R. (2008). Cross-sectional silica exposure measurements at two Zambian copper mines of Nkana and Mufulira. International journal of environmental research and public health, 5(2), 86–90. https:/ /doi.org/10.3390/ijerph5020086.
5. Hayumbu, P., & Robins, T. G. Cross-Sectional Silica Exposure Measurements at Two Zambian Copper Mines of Nkana and Mufulira. International Journal of Environmental Research and Public Health, 5(2), 86-90. https://doi.org/10.3390/ijerph5020086.
It needs moderate editing after effecting all the corrections in the main manuscript.
Author Response
Thank you very much for the valuable and insightful feedback

Reviewer 2 Report
General Comments and Suggestions for the Authors
It is very interesting to get some insight in the occupational situation of mines in Zambia. This is an issue that is neglected too much by the international occupational health and safety community.
The introduction is too long in my opinion. It almost makes the impression of a review paper, which is not intended here as far as I understand. This chapter ranges from page 1 to page 8 of the manuscript, which is about half of the length. It should be significantly shortened. I made some proposals for shortening and moving of text in the detailed comments. It might also help not to repeat the phrase “knowledge, attitudes, work behaviour and practices, compliance with safety standards” (and very similar phrases) about 75 times in the manuscript.
On the other hand relevant information is missing in the introduction. It would be worthwhile to reflect on some of the bigger activities of the last years regarding the evaluation and management of occupational risk factors that use a similar methodology, e.g.:
- Workers’ exposure survey on cancer risk factors in Europe WES, https://osha.europa.eu/en/facts-and-figures/workers-exposure-survey-cancer-risk-factors-europe
- Occupational Integrated Database Exposure Assessment System OccIDEAS, (https://www.occideas.org/)
The WES and OccIDEAS also use surveying as a method for risk assessment of carcinogens. This might be a relevant contribution to this study.
In your definitions of key terms used in the conceptual model (attitudes, behaviour and practice and compliance), chapter 1.3.4. “Model Used in the Study”, page 6, you cite in all cases a couple of authors who contributed to the definitions of these terms. At the end of each of these definition paragraphs you state the definition you choose for your study. Unfortunately, it is not really clear which of the quoted definitions you are referring to in your own definitions. Please try to make this aspect clearer.
In your hypotheses 2 to 3 for your study (chapter 1.3.4. “Model Used in the Study”, pages 7 and 8) you postulate that “Knowledge of the risks and dangers of respirable dust is positively moderated by” different aspects (e.g. compliance, work behaviour, attitude). But is not the other way around also relevant? Could not also e.g. compliance be moderated positively by knowledge? So from my point of view, the question is which aspect influences and which aspect is influenced by another. The manuscript would be improved if you could go into an analysis of cause and effect for your hypotheses.
It would be worthwhile to elaborate why your study is in contradiction to findings of other studies. This is especially true for H1 and H2. But I cannot find any explanation or discussion about these contradictory results. Please add this to your discussion.
Detailed comments
|
Page / Line |
Text |
Comment |
|
1 / 8 |
21School of Built Environment, Copperbelt University, … |
Please check the numbering of this affiliation. Probably it should be “2” and not “21”. |
|
1 / 17 – 18 |
“Statistical package for social sciences (SPSS) was used for descriptives.” |
I would recommend to phrase differently, e.g.: “The statistical package for social sciences (SPSS) was used for the statistical descriptive analysis.” |
|
1 / 20 – 21 |
“The results indicated that of the four (4) hypotheses, two (2) were supported and two (2) were rejected…” |
Why are the numbers additionally set in brackets? |
|
1 / 36 – 38 |
“Risks as a result of physical, emotional, mental and psychosocial factors are associated with the nature of someone’s work and management procedures in an organization. Besides this,” |
Please consider dispensing with this text. It is not the topic of this manuscript. |
|
2 / 45 – 49 |
In 2020, a report by the ILO showed that annually approximately 340 million occupational accidents occur and 160 million victims succumb to work-related illnesses [4]. The ILO (2020) has provided standards on Occupational Health and Safety to guide governments, employers, and workers in ensuring maximum safety at workplaces. |
Please consider dispensing with this text. It is very general information. |
|
2 / 60 – 61 |
Silicosis is a non-preventable incurable occupational respiratory disease caused by inhaling dust particles containing free crystalline silica from their environment. |
Why is silicosis called “non-preventable”? It is actually extremely simple to prevent silicosis. You “just” need to prevent inhalation to silica dust. Please try to rephrase this sentence. |
|
2 / 71 – 74 |
Poor working conditions and the environment within developing countries are substantial concerns because of the lack of preventive measures. Regardless of any industry, workers need to take some time to identify safety and health risks at workplaces, comply and implement appropriate measures for keeping them safe. |
Please consider dispensing with this text. It is very general information. |
|
2 / 90 |
According to World Bank… |
According to the World Bank… |
|
3 / 106 – 110 |
“Another study by Aluko [15] investigated knowledge, attitudes and practices of occupational hazards and safety among healthcare workers revealed that most respondents were knowledgeable about hazards. However, in terms of practice and compliance, only 52% complied with preventive safety precautions as advised by the standard operating procedures, meaning that 48% did not comply.” |
Please consider dispensing with this text. It does not deal with the topic of this manuscript as it is about health care workers but not workers in mines. There are enough examples from the mining industry that you already describe in this chapter. It seems not necessary to expand to other working situations. |
|
3 / 138 – 143 |
“From the literature, several limitations were identified which […] with safety standards with exposure to occupational hazards.” |
Please consider moving these two sentences to the chapter 2. Methods. In my opinion it fits better there. |
|
4 / 158 – 163 |
“In this study, knowledge refers to workers …” |
Please consider moving this sentence to the chapter 2. Methods. In my opinion it fits better there. |
|
4 / 172 |
1.3.1. Health Belief Model (HBM) |
Please check the format of this headline |
|
5 / 208 |
The TTM theorizes that temporal… |
For the other models you chose to start with “The Name of the Model (Abbreviation) …” (see 1.3.1 and 1.3.2). Please consider to harmonize. |
|
5 / 220 |
1.3.4. Model Used in the Study |
Please consider moving this chapter to the chapter 2. Methods. In my opinion it fits better there. |
|
5 / 246 |
… particular behaviour [29] An outcome of a certain … |
Something is wrong with this sentence. Maybe a full stop is missing. Please try to clarify. |
|
6 / 250 – 251 |
Definitions of key terms used in the conceptual model in context with this study are given in Figure 1. |
I do not see that definitions are given in Figure 1. Probably you mean the definitions given below Figure 1. Please clarify this sentence. |
|
8 / 356 |
“… is provided in the supplementary material.” |
The questionnaire is not available to me, therefore it was not part of the review. |
|
8 / 376 – 377 |
“No factors for BP (Work Behaviour and Practice), A (Attitudes), CSS (Compliance and Safety Standards)” |
This is not a sentence. |
|
9 / 394 |
“…and 4.3 – 5 (strongly disagree). |
Should be “strongly agree”. |
|
9 / 395 Table 1 |
Column 1 of the table |
Please improve the design of this table. The first column is hardly readable. |
|
9 / 400 |
“Concerning respirable dust exposure majority of the workers gave a neutral response.” |
Should be: “Concerning respirable dust exposure the majority of the workers gave a neutral response.” |
|
10 / 417 – 419 Figure 2 |
|
The quality of this figure is low. The figure is blurry / out of focus. This clearly needs improvement. I am no familiar with this kind of figure. For my understanding (because of my limited expertise) some more explanation of this figure would be helpful. Within the manuscript this figure is only referenced once on page 8. |
|
14 /527 – 530 |
Even though the hypothesis was not supported in this study there is a need for the management of the mining site to seriously undertake appropriate measures in educating workers on the risks and dangers of respirable dust exposure because this can reduce the chances of contracting respiratory diseases. |
It may be, that I do not understand this sentence correctly. But my understanding is that “knowledge” does not influence (or even reduce) exposure according to your results. But you still advocate to improve the knowledge because it can reduce. This does not make sense in my opinion.
You should at least add something along the lines “other studies showed this” or similar. |
|
|
|
|
|
15 / 625 – 626 |
Supplementary Materials: The questionnaire that was used to collect data from the miners in this article has been included as supplementary material. |
The questionnaire is not available to me, therefore it was not part of the review. |
|
17 / 715 – 717 |
Reference 42. Beth AA. … |
Please check formatting of this reference (should not be bold). |
I will not comment about the English language and style (with some small exemptions). At least some of the authors seem to be native speakers and well established scientists in their field. I am not a native speaker and will therefore not comment on their use of the English language.
Author Response
Thank you so much for the valuable and insightful feedback
